# New Insights into the Antimicrobial Potential of *Polyalthia longifolia*—Antibiofilm Activity and Synergistic Effect in Combination with Penicillin against *Staphylococcus aureus*

**DOI:** 10.3390/microorganisms10101943

**Published:** 2022-09-30

**Authors:** Mihaela Savu, Marguerite Kamdem Simo, Gabriel Xavier Fopokam, Stefan Mihaita Olaru, Oana Cioanca, Fabrice Fekam Boyom, Marius Stefan

**Affiliations:** 1Department of Biology, Faculty of Biology, The Alexandru Ioan Cuza University of Iasi, Bd. Carol I, No. 11, 700506 Iasi, Romania; 2Department of Biological Sciences, Faculty of Sciences, University of Maroua, Maroua P.O. Box 814, Cameroon; 3Department of Biochemistry, Faculty of Sciences, University of Yaoundé I, Yaoundé P.O. Box 812, Cameroon; 4Faculty of Pharmacy, “Grigore T. Popa” University of Medicine and Pharmacy, 16 Universitatii Street, 700115 Iasi, Romania

**Keywords:** *Polyalthia longifolia* leaf extracts, chemical composition, bactericidal effect, membrane impairment, antibiofilm activity, synergistic effect

## Abstract

Widespread antibiotic resistance has led to the urgent need for the identification of new antimicrobials. Plants are considered a valuable potential resource for new effective antimicrobial compounds. Therefore, in the present study, we focused on the antimicrobial activity of *Polyalthia longifolia* plants harvested from Cameroon using the minimum inhibitory concentration (MIC), minimum bactericidal concentration (MBC), and time-kill assays. The mechanism of action was investigated by employing fluorescence and scanning electron microscopy. The anti-*Staphylococcus aureus* activity was studied using biofilm inhibition and checkerboard assays. Our results revealed that the tested extracts possess important antimicrobial activities, notably against Gram positive bacteria (MICs as low as 0.039 mg/mL). *P. longifolia* leaf extracts exhibited a significant bactericidal effect, with a total kill effect recorded after only 2 h of exposure at concentrations equivalent to MBC (0.078 and 0.156 mg/mL). The extracts showed a synergistic antibacterial activity in combination with penicillin against a MRSA clinical isolate and significantly inhibited *S. aureus* biofilm formation. The mechanism of action is related to the impairment of cell membrane integrity and cell lysis. All these findings suggest that *P. longifolia* could be an important source of reliable compounds used to develop new antimicrobials.

## 1. Introduction

Antimicrobial resistance (AMR) is considered a major public health concern. Estimates from the ECDC and WHO show that each year in the European Union alone, more than 670,000 infections are due to bacteria resistant to antibiotics, and approximately 33,000 people die as a direct consequence of these infections [1]. AMR is rising to dangerously high levels all around the world, particularly in African countries such as Cameroon, in direct relation to the indiscriminate use of antibiotics [2]. Despite limited laboratory capacity to monitor AMR, available data indicate that the African region shares the worldwide trend of increasing drug resistance. Significant resistance has been reported for several bacteria that are likely to be transmissible not only in hospitals, but also in the community [3]. AMR leads to disastrous financial consequences, including extremely high medical costs due to an increase in hospital admissions, drug usage, and increased mortality [4].

The pathogenic microorganisms that are highly resistant to antibiotics include bacteria (such as *Staphylococcus aureus*, *Enterococcus* spp., *Enterobacteriaceae*, and *Pseudomonas aeruginosa*) and fungi (e.g., *Candida albicans*) [5]. The infections caused by these pathogens are harder to treat than those caused by non-resistant microorganisms. In addition, the misuse and abuse of traditional antibiotics have reduced their efficacy, posing not only a health issue, but also a global development problem [6]. Consequently, antimicrobial drugs have become less effective, or even ineffective, rapidly outpacing available treatment options [6]. Therefore, it is crucial to preserve the efficacy of existing drugs using measures to minimize the development and spread of AMR, while efforts to develop new treatment options proceed [7].

Multiple approaches are being employed in the discovery of new antibiotics. One of these is represented by the use of natural products, which remain a promising source of new efficient antimicrobial agents [8]. For centuries, medicinal plants such as the *Annonaceae* species have been used in traditional medicines to cure various microbial diseases [9]. *Polyalthia longifolia* (false ashoka tree) is used in traditional medicine for the treatment of fever, skin diseases, diabetes, hypertension, and helminthiasis [10]. A literature survey revealed that aqueous and methanolic extracts of *P. longifolia* leaves also showed important antimicrobial properties [11,12,13,14].

To the best of the author’s knowledge, no studies have been carried out on the antimicrobial properties of *P. longifolia* leaf and stem extracts from wild plants of the Cameroon region. Although various papers have confirmed the antimicrobial activities of the crude extracts and isolated compounds from *P. longifolia*, their mechanism of action is not fully understood. In addition, less is known about the chemical composition (most studies contain data regarding the volatile fraction and preliminary screening of the bioactive metabolites, without the indication of any polyphenolic compounds), antibiofilm activity, and synergistic effect with penicillin against pathogenic microbial strains. Therefore, in the present study, we address not only the antibacterial and antifungal properties of *P. longifolia* extracts, but also their chemical composition (identification and quantification by RP-UPLC-PDA techniques), mechanism of action, and the effect in combination with penicillin against a MRSA strain, as well as the activity against biofilm formation—an important *S. aureus* virulence factor.

## 2. Materials and Methods

### 2.1. Plant Material

*Polyalthia longifolia* plants were harvested in June and September 2021 in Yaoundé (Centre Region, Cameroon), identified at the National Herbarium of Cameroon by comparison with existing specimens, and recorded under the code number 67474HNC.

### 2.2. Extract Preparation

The aerial parts of the plants (leaves and stem) were chopped and dried at room temperature in a place protected from direct sunlight. Dried organs were ground, and 30 g of each resulting powder was macerated in 0.5 L of solvent (water, methanol, or ethanol) for 72 h at room temperature. The resulting mixture was filtered through filter paper (Whatman No. 3), and the solvent was completely evaporated. The dry residue obtained was stored at 4 °C for later use.

### 2.3. Phytochemical Analysis

The chemical composition of *Polyalthia longifolia* leaf extracts (PlLm and PlLe) was assessed using ultra-high performance liquid chromatography (UHPLC) analysis, as previously described [15]. The system included a Thermo UltiMate 3000 chromatograph with a quaternary pump, an autosampler, a Phenomenex Luna Omega Polar C18 column (100 A, 150 mm × 4.6 mm), and a PDA (multidiode array detector). Five lengths ranging between 245 nm and 520 nm were simultaneously registered for the detection of the compounds. The identification of the compounds from the two extracts was based on the retention time, UV spectra comparison with various standards (Sigma Aldrich, Darmstadt, Germany), and the Chromeleon, Nist, and Wiley data libraries. The available standards were catechin, epicatechin, caffeic acid, rosmarinic acid, chlorogenic acid, luteolin, apigenin, quercetin-3-arabinoside, apigenin-7-O-glucoside, luteolin-7-O-glucoside, loganin, and harpagide (all HPLC quality, Sigma, Germany). The standard stock solutions, in concentrations varying from 0.114–0.195 mg/mL, were analyzed in identical conditions with the samples, and the calibration curves were obtained with a standards deviation of 0.009 and a correlation coefficient of 0.9989. The limit of detection (LOD) and the limit of quantification (LOQ) of epicatechin and chlorogenic acid were calculated at 280 ng/mL and 145 ng/mL. Chromeleon 7.2 v.12 software was used for the integration of the peaks.

### 2.4. Microbial Strains

Test microorganisms—*Staphylococcus aureus* ATCC 25923, *Escherichia coli* ATCC 25922, and *Candida albicans* P37037—strains were provided by the culture collection of the Microbiology Laboratory, Alexandru Ioan Cuza University of Iasi. The methicillin-resistant *Staphylococcus aureus* (MRSA) isolate was kindly provided by Med. biol. PhD Simona Matiut from Praxis Clinical Laboratory, Iasi, Romania, and included in the same microbial culture collection mentioned above, with the following accession number: prx-MRSA-2018.

All strains were stored in 15% glycerol stocks at −80 °C. Prior to experiments, *S. aureus* and *E. coli* strains were transferred to Mueller–Hinton agar (MHA, Liofilchem, Roseto degli Abruzzi, Italy) and *C. albicans* on Sabouraud dextrose agar (SDA, Carl Roth, Karlsruhe, Germany) and incubated at 37 °C. Subsequently, 10 mL of Mueller–Hinton broth (MHB, Scharlau, Barcelona, Spain) and 15 mL of Sabouraud dextrose broth (SDB, Carl Roth, Germany) were inoculated with one representative colony of each test organism taken from MHA or SDA, cultured overnight (37 °C, 190 rpm and 130 rpm, respectively) and used as source of inoculum for further experiments.

### 2.5. Assessment of Antimicrobial Activity

#### 2.5.1. Determination of Minimum Inhibitory Concentration (MIC) and Minimum Bactericidal/Fungicidal Concentration (MBC/MFC)

The MIC was determined by the broth microdilution method, as previously described [16,17]. A concentration range of *P. longifolia* extracts between 0.0097 and 20 mg/mL was tested, with dimethyl sulfoxide (DMSO, Merck, Darmstadt, Germany) as solvent. DMSO was used as a control at concentrations ranging from 12.5 to 0.006% (*v*/*v*). Ampicillin, kanamycin, and nystatin were used as reference antimicrobials.

The inoculum was added to each well of a microplate (final cell densities were approximately 1.5 × 10^6^ CFU/mL for bacterial inoculum and 2.5 × 10^3^ CFU/mL for fungal inoculum), except the wells containing only MHB (used for *S. aureus* and *E. coli* strains) or SDB (used for *C. albicans* strain) medium considered as blank. Growth control was represented by inoculum and MHB or SDB medium alone. MIC was expressed as the lowest concentration showing no visible growth. To evaluate MBC or MFC, a volume of 15 μL was taken from each well with no visible growth, inoculated on MHA or SDA plates, and incubated at 37 °C. The MBC/MFC was considered the lowest concentration at which bacterial and fungal cells failed to grow after plating onto MHA or SDA plates.

#### 2.5.2. Microbial Growth Analysis

The effect of the *P. longifolia* extracts PlLm and PlLe on *S. aureus* ATCC 25923 growth was assessed, as previously described [17]. A volume of 200 μL from a 0.5 McFarland cell suspension was used to inoculate 20 mL MHB (final cell density was approximately 1.5 × 10^6^ CFU/mL). The medium was supplemented with PlLm and PlLe extracts to obtain final concentrations equivalent to ½ MIC, MIC, and 2 × MIC. The control was represented by inoculated MHB medium supplemented with DMSO at appropriate concentrations, without PlLm and PlLe extracts. All flasks were incubated on an orbital shaker (190 rpm) at 37 °C for 12 h. Samples were taken every hour and growth monitored by measuring the optical density at 600 nm (OD_600_), using a Beckman Coulter DU 730 spectrophotometer.

#### 2.5.3. Time-Kill Assay

Time-kill curves were obtained using *S. aureus* ATCC 25923 as a test microorganism. A volume of 100 µL from bacterial suspensions (approximately 0.5 McFarland) was added to 10 mL PBS supplemented with different concentrations of PlLm and PlLe extracts and incubated at 37 °C and 190 rpm. The same procedure was followed to prepare the controls, for which plant extracts were replaced with DMSO. Aliquots were removed at specific times (each hour up to 12 h, and then at 24 and 48 h), serially diluted, and plated on MHA. After 24 h of incubation, the viable cell number reported as CFU per mL was calculated by colony counting and transformed into log10 values. Bactericidal activity was defined as a ≥3 log10 reduction in the total CFU/mL from the original inoculum. Time-kill curves were constructed by plotting mean colony counts versus time [18].

### 2.6. Evaluation of Membrane Integrity

A membrane permeability test was performed to assess the effect of PlLm and PlLe extracts on *S. aureus* ATCC 25923 cell membrane integrity [17]. Exponential-phase cell suspensions (final cell density approximately 1 × 10^8^ CFU/mL) were incubated at 37 °C on an orbital shaker (190 rpm) in the presence of tested extracts at concentrations equivalent to ½ MIC values with DMSO as a control. Aliquots of treated and untreated cells were taken at 0.25, 0.5, 1, 1.5, 2, 3, 4, and 5 h and stained with propidium iodide (PI, Carl Roth, Germany) for 15 min in the dark. The fluorescent cells were examined and counted using a Leica DM1000 LED fluorescence microscope and an I3 blue excitation range filter cube (BP 450 ± 490 nm band-pass filter). The ratio between fluorescent cells and total cells was calculated as a percentage using at least five random images captured per sample.

### 2.7. Scanning Electron Microscopy (SEM)

A 5 mL volume of a *S. aureus* ATCC 25923 cell suspension (approximately 1 × 10^8^ CFU/mL) was incubated for 4 h in the presence of PlLm extract (final concentrations equivalent to MIC and 2 × MIC values). DMSO served as a control. Samples (untreated and treated *S. aureus* cells) were prepared for SEM analysis, following the protocol previously described [17], and examined with a Tescan Vega II SBH microscope using the secondary electron detector at an acceleration voltage of 30 kV.

### 2.8. In Vitro Biofilm Formation

The *S. aureus* ATCC 25923 strain was used to develop *in vitro* biofilms in flat-bottomed sterile 96-well microplates with lids (Becton Dickinson, Franklin Lakes, NJ, USA) [17]. In each well, a volume of 100 μL Luria Bertani medium (LB) was inoculated with 100 μL bacterial suspensions (final density was approximately 1 × 10^6^ CFU/mL). Only in the wells considered as samples, the medium was supplemented with PlLm and PlLe extracts to reach final concentrations ranging from 0.0097 to 0.312 mg/mL. The wells containing only inoculated LB supplemented with DMSO served as the control. The plates were incubated at 37 °C for 48 h. The developed biofilm was assessed using crystal violet staining [16]. Biofilm formation in the presence of PlLm and PlLe was expressed as a percentage of the control biofilm formed in the absence of tested *P. longifolia* extracts (considered as 100%).

### 2.9. Checkerboard Assay

The effect of PlLm and PlLe extracts in combination with penicillin (Carl Roth, Germany) against a MRSA clinical isolate was assessed using the checkerboard microdilution method, as previously described [16], with minor modifications. All dilutions were performed in MHB medium (50 μL per well). Different combinations of concentrations were obtained by transferring penicillin dilutions (50 μL) to the plate containing plant extracts. The inoculum (100 μL of bacterial suspension, prepared as presented above) was added to each microplate well (final cell density was approximately 1.5 × 10^6^ CFU/mL) and incubated at 37 °C for 24 h. MHB medium and inoculum served as the control. Bacterial growth was assessed using resazurin [19]. The fractional inhibitory concentration index (FICI) was calculated for each combination to evaluate the synergistic effect, as previously described [16]. The FICI results for each combination were defined as synergy (≤0.5), additivity (0.5 < FICI ≤ 1), indifference (1 < FICI ≤ 4), and antagonism (>4).

### 2.10. Statistical Analysis

All experiments were performed in triplicate. The statistical evaluation of the results was carried out by Šidák’s multiple comparisons test (time-kill assay, evaluation of membrane integrity, *in vitro* biofilm formation) and Dunnett’s multiple comparisons test (microbial growth analysis) using GraphPad Prism 9 software. Differences between groups were considered significant when *p* < 0.05. The data are presented as mean (*n* = 3) ± S.E.M.

## 3. Results

### 3.1. Antimicrobial Activity of P. longifolia Extracts

The results of the antibacterial and antifungal activities are presented in Table 1. *P. longifolia* extracts showed a promising activity against *S. aureus* (MICs as low as 0.039 mg/mL) and were moderately active against *E. coli* (MICs of 5 and 10 mg/mL). The lowest MBC values were recorded for PlLm and PlLe extracts against *S. aureus* (0.078 mg/mL and 0.156 mg/L, respectively). The same extracts also showed a good antibacterial activity against a MRSA clinical isolate, with MIC values of 0.312 and 0.624 mg/mL, respectively, and MBC value of 10 mg/mL. Concerning the antifungal activity, the lowest MIC or MFC recorded value was 5 mg/L for several *P. longifolia* extracts.

Based on lower MIC and MBC values, extracts PlLm and PlLe were selected for further phytochemical analysis and antibacterial tests against *S. aureus* strains.

### 3.2. The Chemical Composition of PlLm and PlLe Extracts

The chemical analysis of the extracts was performed using a fast liquid chromatography method. The main compounds identified in the analyzed extracts (flavones, polyphenols, and tannins) are presented in Table 2. No significant differences were recorded between the methanolic and ethanolic *P. longifolia* leaf extract compositions, as can be seen in the chromatograms depicted in Figure 1.

#### 3.2.1. *P. longifolia* Extracts Inhibited *S. aureus* Growth in a Dose and Time Dependent Manner

The MIC values (0.039 mg/mL and 0.078 mg/L) were used as references to evaluate the effect of PlLm and PlLe extracts on *S. aureus* growth over time. The tested extracts induced a significant growth delay up to 6 and 7 h, respectively, compared to the control at concentrations equivalent to ½ MIC (Figure 2). The analysis of the growth dynamics curves also revealed that increasing concentrations of the tested extracts induced a progressive inhibition of the bacterial growth. Concentrations equivalent to MIC and 2 × MIC suppressed the growth of *S. aureus* cells within the time span of the experiments, showing an important bacteriostatic activity.

#### 3.2.2. *P. longifolia* Leaf Extracts Have a Potent Bactericidal Activity against *S. aureus* Cells

A total kill effect (no viable cells) was recorded when bacterial cells were exposed to both *P. longifolia* tested extracts at concentrations equivalent to 2 × MIC after only 2 h of incubation (Figure 3). We must emphasize that no viable *S. aureus* cells were detected up to 48 h of exposure.

### 3.3. Mode of Action

The PlLm and PlLe extracts mechanism of activity against *S. aureus* cells was investigated using fluorescence microscopy and SEM.

#### 3.3.1. PlLm and PlLe Extracts Impair the Integrity of the Cellular Membrane

Fluorescence microscopy was employed to assess the penetration of PI into *S. aureus* cells with damaged membranes after exposure to the tested extracts at concentrations equivalent to ½ MIC (Figure 4). Control cells showed negligible levels of fluorescence throughout the entire experimental period, proving the inability of PI to penetrate viable cells with intact cytoplasmic membranes [20]. Following exposure to the PlLm extract, the number of fluorescent cells significantly increased compared with the control, starting at 15 min of incubation (*p* = 0.0001). After 2 h of incubation, more than 71.49% of cells exposed to PlLm were fluorescent, most likely due to significant membrane damage (Figure 4b). Similar results were also obtained for the PlLe extract (data not shown).

#### 3.3.2. Irreversible Cell Morphological Damage Is Induced by PlLm Extract

Scanning electron microscopy analysis clearly revealed severe alteration of the cell morphology after exposure to PlLm extract at concentrations equivalent to MIC and 2 × MIC (Figure 5). *S. aureus* exposed cells appeared agglutinated, deformed, clustered, and collapsed (Figure 5b,c). Moreover, the cell shrinkage was evidenced, along with cellular debris, most likely resulting from the disintegration of the cells. Control cells showed a normal morphology, with a smooth surface and clear boundaries (Figure 5a).

### 3.4. Bacterial Biofilm Formation Was Inhibited by PlLm and PlLe Extracts

Different concentrations of *P. longifolia* extracts induced a significant inhibitory effect on bacterial biofilm formation (Figure 6). The highest biofilm inhibition percentages were recorded for the PlLm and PlLe extracts at concentrations equivalent to MIC 99.5% (0.039 mg/mL) and 97.47% (0.078 mg/mL), respectively. We must emphasize that the antibiofilm activity was also recorded for subinhibitory concentrations such as ½ MIC and ¼ MIC (Figure 6), although the last concentration was not significantly different compared with the control for PlLm extract.

### 3.5. Effect of P. longifolia Extracts in Combination with Penicillin against a MRSA Strain

The checkerboard method was used to investigate whether the combination of PlLm and PlLe extracts with penicillin provided a synergistic effect against a MRSA clinical isolate. The FICI that defines synergy between *P. longifolia* extracts and penicillin is shown in Table 3. Penicillin showed synergistic effects with both extracts (FICI values: 0.18–0.28). We must point out that in combination with PlLm and PlLe extracts, penicillin showed noticeably improved antibacterial activity, with 33–66-fold reduced MIC values.

## 4. Discussion

The widespread of antibiotic resistance phenomenon, along with the exhaustion of the antibiotic pipeline, has led to the urgent need for the identification of new antimicrobial molecules. Plant chemical biodiversity is a valuable potential resource for new effective antimicrobial drugs [21]. Compared to synthetic molecules, natural products are better tolerated by living organisms and are more cost-effective [22]. In this context, we hypothesize that *Polyalthia longifolia* plants could be a reliable source of new antimicrobials. *P. longifolia* extracts have been described in the past to possess prominent antibacterial properties [23,24,25]. However, their chemical composition and mechanisms of action are not fully understood. Therefore, in the present study we have focused not only on the antimicrobial activity of *P. longifolia* extracts, but also on the chemical composition, mechanism of action, antibiofilm activity, and the effect in combination with penicillin against a methicillin-resistant *Staphylococcus aureus* strain.

*P. longifolia* extracts exhibited a promising antimicrobial activity against all bacterial and fungal strains tested *in vitro*. However, the recorded antimicrobial effect was lower compared with the antimicrobials used as a reference in our study. The literature survey revealed that *P. longifolia* leaf and stem extracts tested in our study showed a similar or better antibacterial activity compared with previously presented data (MICs ranging from 0.0125 to 50 mg/mL) [13,25]. Additionally, *P. longifolia* extracts presented a more pronounced antifungal activity (MICs of 5 or 10 mg/mL) compared with crude leaf extracts previously tested (MIC = 25 mg/mL) [25]. Nevertheless, some authors reported lower MIC values against *C. albicans*—0.25 mg/mL—than those presented in our paper [13]. PlLm and PlLe showed the lowest MBC values of all the tested extracts: 0.078 and 0.156 mg/mL respectively. The recorded MBC values are comparable to some previous reported data for ethanolic extracts (MBC range between 0.02 and 2.5 mg/mL) [26] or diterpenoid isolated from *P. longifolia* var. pendula (MBC = 0.185 mg/mL) [27], although there is limited existing literature on the bactericidal activity of *P. longifolia* extracts. No significant differences were registered between the antimicrobial activity of methanolic and ethanolic leaf and stem extracts, most likely due to their similar chemical composition. However, the MIC and MBC values recorded for water extracts against *S. aureus* ATCC 25923 were significantly higher compared with methanolic and ethanolic extracts, and this could be related to the different solvent used for extraction and therefore, a possibly different chemical composition.

The antibacterial effect of methanolic and ethanolic extracts in terms of MIC and MBC values was significantly higher against *S. aureus* ATCC 25923 compared with *E. coli* ATCC 25922. The lowest MIC value (0.039 mg/mL) was recorded for the *P. longifolia* leaf extract against the *S. aureus* strain when methanol was used for extraction. Different susceptibilities of Gram positive and Gram negative strains to PlLm and PlLe are most likely attributed to different cell wall structures, which could be explained by the chemical composition of the tested extracts. Thus, compounds such as catechin (identified in both PlLm and PlLe extracts) can directly bind to peptidoglycan (an essential component of the cell wall in Gram-positive bacteria), interfering with its biosynthesis [28] and explaining the higher susceptibility of *S. aureus* to PlLm and PlLe. Although *E. coli* cell walls contain several layers of peptidoglycan, the presence of the outer membrane provides protection against different antimicrobials, explaining the milder effect of PlLm and PlLe extracts [28].

Based on the lower MIC and MBC values against *S. aureus*, PlLm and PlLe extracts were selected for further tests regarding antibacterial activity, phytochemical analysis, and mechanism of action.

Both PlLm and PlLe extracts were active against a MRSA clinical isolate, with MIC values of 0.312 and 0.624 mg/mL, respectively. Methicillin-resistant *Staphylococcus aureus* is an ESKAPE pathogen, with high clinical importance and which is responsible for various major life-threatening nosocomial infections [29]. Most ESKAPE pathogens are multidrug-resistant strains, which is one of the greatest challenges in clinical practice [30]. Therefore, we consider PlLm and PlLe antibacterial activity very attractive, suggesting the potential of developing new drugs to overcome MRSA-related infections. Our conclusion is supported by a comparative literature survey that revealed the same potential of *P. longifolia* extracts against clinical isolates of MRSA [11,29,31]. The important antibacterial activity of *P. longifolia* leaf extracts could also be explained in relation to its chemical composition. Thus, some biologically active compounds such as catechin, epicatechin, caffeic and rosmarinic acids, apigenin, or quercetin, well known for their antimicrobial properties [32,33,34,35,36,37,38,39], were identified in the composition of both tested extracts.

The antibacterial activity of PlLm and PlLe extracts was also investigated using growth kinetic studies. Our results revealed that concentrations equivalent to ½ MIC induced a significant growth delay compared with the unexposed control cells up to 6 and 7 h of incubation in the presence of PlLm and PlLe extracts, respectively. Increasing the concentration up to MIC and 2 × MIC progressively inhibited *S. aureus* growth over time, denoting a significant dose-dependent inhibitory effect. We must emphasize that no turbidity was recorded for all samples exposed to concentrations equivalent to MIC and 2 × MIC during the 12 h experiments, implying important bacteriostatic activity.

The lack of *S. aureus* cell viability during exposure to PlLm and PlLe extracts was evidenced using a time-kill kinetics assay. The results confirmed the potent bactericidal activity of both extracts during the 48 h incubation in the presence of MIC and 2 × MIC. Our data showed that a total kill effect (no viable cells) was recorded starting at only 2 h after exposure at 0.078 and 0.156 mg/mL (equivalent to MBC), suggesting a significant bactericidal potential.

*P. longifolia* is a recognized remedy in traditional Indian, Tibetan, Chinese, Malaysian, and African medicine. All parts of the tree are used to treat various disorders [40]. The literature abounds with data regarding the terpene fraction found in the leaves of *P. longifolia*, for which several studies evaluated the antimicrobial and antifungal activities. However, most studies contain data regarding the volatile fractions and preliminary screening of the bioactive metabolites, without the indication of other types of secondary metabolites, such as polyphenolic compounds, that may influence the biological properties [41]. Moreover, the majority of the extracts were obtained using non-polar solvents (petroleum ether, n-hexane, ethyl acetate, methanol) when clerodane diterpenes were isolated [42,43]. However, our study focused on the identification of the polyphenols present in the investigated extracts. The chemical composition of the PlLm and PlLe samples is concordant with the solubility and polarity of both solvents (methanol and ethanol). Most of the identified compounds have an increased extractability in alcohol, but minor affinities allow the higher polarity compounds to be extracted in ethanol, whereas larger quantities of aglycons and fewer polar substances are isolated with methanol. As indicated by the chromatograms, our extracts are richer in catechins, and gallic acid derivatives were mainly identified in the ethanolic extract [41]. Larger quantities of rosmarinic acid were also noted for both samples. The similarity in the chemical profile is given by the identical origin of the plant material (leaves of *Polyalthia longifolia*), but the differences regarding the ratio between the compounds is due to the variable affinity of each substance in the chosen solvents (methanol or ethanol). As compared to other studies, the recorded differences may be explained by the different origin of the plants, as most of the previously reported data were obtained from studies conducted on plants collected from India, Malaysia, or Nigeria [24,40,44,45,46]. It is known that the pedo-climatic conditions significantly impact the plants’ biosynthetic capacity, inducing changes in the chemical profile of all secondary metabolites. Such metabolites are produced as a mechanism to adapt to the environmental changes, polyphenols being known as photoprotectors against radiations and acting as antioxidants to prevent the formation free radicals, thus reducing cell membrane degradation [47]. Polyphenols are also synthetized in plants for protection against various pathogens. Catechins, apigenin, galangin, and naringenin are among the most studied polyphenols because of their antimicrobial potential. Due to their chemical structure, the mechanism of action of conjugated double bonds and free hydroxyl moieties is related to their capacity to bind proteins in complexes through hydrogen and covalent bonding, as well as their hydrophobic effects. Although the mechanisms are complex, research has shown that polyphenols can interact with the cell wall synthesis, inhibit certain bacterial enzymes, and disrupt DNA and RNA synthesis in bacteria [48,49,50,51].

The effect of PlLm and PlLe extracts on the cell membrane was investigated using fluorescence microscopy. Exponential-phase *S. aureus* cells were treated with concentrations equivalent to ½ MIC and stained with PI to visualize dead or injured cells. The cells become fluorescent as PI penetrates the bacterial membrane when it is damaged [52]. Our results revealed a progressive increase in the fluorescent cells number over time, starting at 15 min of incubation in the presence of the tested extracts. After 5 h, more than 90% of cells exposed to PlLm and PlLe extracts at concentrations equivalent to ½ MIC were fluorescent, suggesting significantly damage to the bacterial cell membrane integrity. Damaged cell membranes most frequently lead to cell lysis. Therefore, SEM was employed to determine morphological damages and the possible cell lysis induced by *P. longifolia* leaf extract exposure. SEM image analysis showed severe morphological alterations, extensive cell shrinkage, and agglutination, with cellular debris most likely resulting from cell lysis. All these results clearly indicate that the *P. longifolia* leaf extracts target the cellular membrane, inducing membrane disruption and cell lysis. Our hypothesis concerning the mechanism of action is supported by previous reports showing that compounds such as catechin, identified in both PlLm and PlLe extracts, exert an antibacterial effect by intercalating into the lipid bilayer, leading to lateral expansion and membrane disruption [33]. In addition, other compounds detected in *P. longifolia* leaf extracts, such as apigenin, rosmarinic acid, and caffeic acid, also target the cellular membrane, as previously revealed [37,38,53]. On the other hand, catechin is known to interfere with the peptidoglycan biosynthesis [28]. Therefore, we may presume that PlLm and PlLe extracts could act on multiple cellular targets—membrane and cell wall—but further studies are necessary to test this hypothesis.

The formation of biofilms is a major virulence factor influencing *S. aureus* pathogenesis [52]. Due to this characteristic, *S. aureus* is a leading cause of human infection [54]. Our results showed that both PlLm and PlLe extracts significantly inhibited biofilm formation, in a dose-dependent manner. Moreover, the inhibition effect was recorded for sub-MIC levels (½ MIC and ¼ MIC), denoting an important antibiofilm activity. The inhibition of biofilm formation could be the result of the EPS structural components degradation or the inhibition of the sessile cells in the DNA and RNA content induced by compounds such as catechin [55]. The antibiofilm activity could also be explained by the presence of rosmarinic and caffeic acids or quercetin, compounds previously reported for their antibiofilm potential [33,38,56,57].

Biofilm formation also represents one of the main mechanisms involved in antibiotic resistance. Several strategies have been proposed to address this major problem in modern medicine. One of these is related to the use of synergistic combinations of new compounds with traditional antibiotics used in therapy to which *S. aureus* strains have already become resistant. In our study, which involved a MRSA clinical isolate, PlLm and PlLe extracts significantly potentiated the efficacy of penicillin against MRSA by a range of 33–66-fold, suggesting an important synergistic antibacterial activity. The synergism between *P. longifolia* extracts or isolated compounds (catechin, epicatechin, caffeic and rosmarinic acids) and different antibiotics, such as ampicillin [58], oxacillin, tetracycline, daptomycin, linezolid [59], erythromycin, clindamycin, cefoxitin, and vancomycin [60,61] was previously reported. However, less is known about the synergistic interaction of *P. longifolia* extracts with penicillin against MRSA strains. We may presume that this synergistic effect could be explained by the different mechanism of action of the molecules involved in combination, i.e., penicillin interferes with the peptidoglycan biosynthesis, and therefore targets the formation of a new cell wall, while PlLm and PlLe extracts target the cellular membrane by impairing its integrity. Our data support the potential of *P. longifolia* extracts as a novel antibacterial tool against multidrug-resistant strains.

## 5. Conclusions

Our results have shown that extracts obtained from *Polyalthia longifolia* plants harvested from Cameroon possess important antimicrobial activities, especially against Gram positive bacteria such as *S. aureus*. *P. longifolia* leaf extracts expressed synergistic antibacterial activity in combination with penicillin against a MRSA clinical isolate, inhibiting biofilm formation. The chemical composition of *P. longifolia* extracts (especially the presence of catechins and rosmarinic acid) explains the antimicrobial potential, as well as the possibility of interaction with antimicrobial agents. The mechanism of action is related to the cell agglutination and impairment of cell membrane integrity, leading to cell lysis. All these findings suggest that *P. longifolia* leaf extracts could be an important source of reliable compounds used to develop new antimicrobials to fight antibiotic resistance.

## Figures and Tables

**Figure 1 microorganisms-10-01943-f001:**
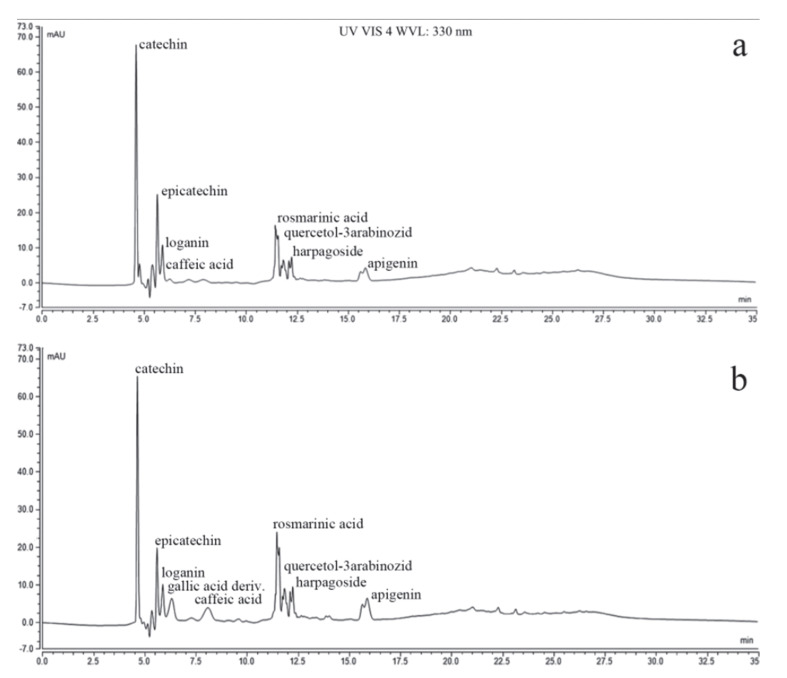
Chromatograms of *Polyalthia longifolia* leaf extracts PlLm (**a**) and PlLe (**b**).

**Figure 2 microorganisms-10-01943-f002:**
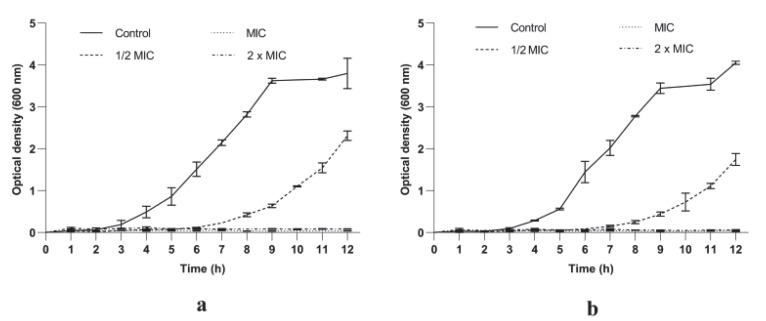
Growth kinetics of *S. aureus* ATCC 25923 treated with extracts PlLm (**a**) and PlLe (**b**) at different concentrations. Bars indicate standard deviations.

**Figure 3 microorganisms-10-01943-f003:**
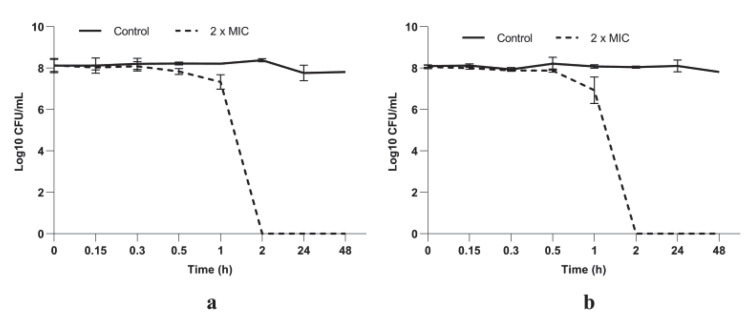
Time-kill curves of (**a**) PlLm and (**b**) PlLe extracts against *S. aureus* ATCC 25923. Bars indicate standard deviations.

**Figure 4 microorganisms-10-01943-f004:**
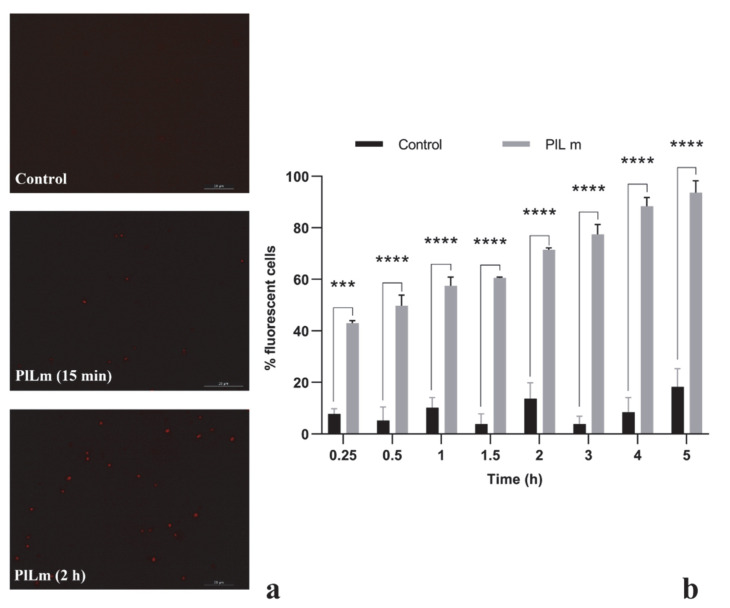
Effect of *P. longifolia* methanolic leaf extract exposure on *S. aureus* cell membrane integrity. Exponential-phase cells were treated with DMSO (control), PlLm extract (concentration equivalent to ½ MIC—samples), and stained with propidium iodide. Negligible levels of fluorescence were detected in the control after 3 h of incubation. Red fluorescent cells were detected in samples after 15 min and 2 h of incubation with PlLm, indicating membrane damage (**a**). The number of fluorescent cells significantly increased over time in the PlLm exposed samples (**b**). Values are the mean ± SEM. Bars indicate standard deviations. Asterisks denote a significant difference vs. Control (*** = *p* = 0.0001; **** = *p* < 0.0001).

**Figure 5 microorganisms-10-01943-f005:**
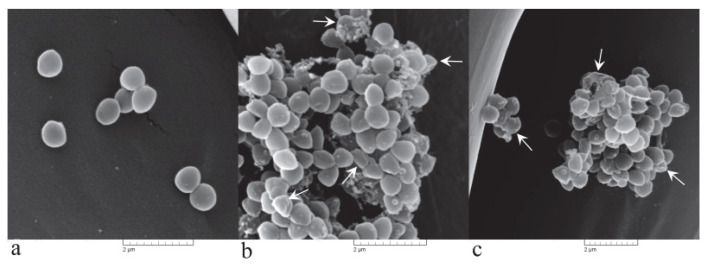
SEM photomicrographs showing the effects of PlLm extract on *S. aureus* cell morphology: (**a**) control; (**b**,**c**) cells exposed for 4 h to MIC and 2 × MIC, respectively. White arrows indicate irreversible morphological damage of treated bacterial cells and cellular debris.

**Figure 6 microorganisms-10-01943-f006:**
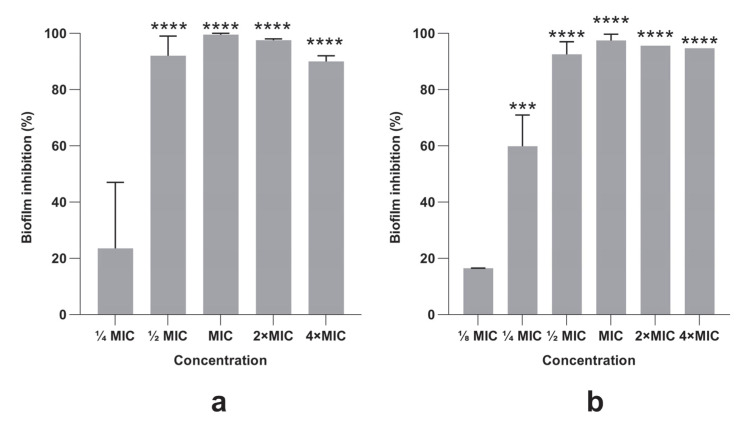
Anti-biofilm activity of (**a**) PlLm and (**b**) PlLe extracts at different concentrations. Values are the means of at least three replicates. Bars indicate standard deviations. Asterisks represent a significant difference (*p* < 0.05) vs. control (*** = *p* = 0.0001; **** = *p* < 0.0001).

**Table 1 microorganisms-10-01943-t001:** Minimum inhibitory concentration and minimum bactericidal/fungicidal concentration of *P. longifolia* extracts (mg/mL).

Extract	*S. aureus* ATCC 25923		*E. coli* ATCC 25922		*C. albicans* P37037	
	MIC	MBC	MIC	MBC	MIC	MFC
PlLm	0.039	0.078	10	20	5	10
PlLe	0.078	0.156	10	20	10	10
PlLw	5	10	5	10	5	5
PlSm	0.310	20	10	20	5	10
PlSe	0.152	20	10	20	5	10
PlSw	5	20	5	20	5	5
Ampicillin	0.00781	-	0.00781	-	-	-
Kanamycin	0.00195	-	0.00195	-	-	-
Nystatin	-	-	-	-	0.03	-

MIC = minimum inhibitory concentration; MBC = minimum bactericidal concentration; MFC = minimum fungicidal concentration; PlLm—leaves methanolic extract; PlLe—leaves ethanolic extract; PlLw—leaves water extract; PlSm—stem methanolic extract; PlSe—stem ethanolic extract; PlSw—stem water extract. The values are the mean of three replicates.

**Table 2 microorganisms-10-01943-t002:** Phytochemical analysis of *Polyalthia longifolia* leaf extracts.

Compound	Extracts	
	PlLm	PlLe
**Flavonoids (mg/mL *)**		
1. Catechin	42.8096 ± 0.0213	41.8657 ± 0.0511
2. Epicatechin	22.7416 ± 0.0110	19.5180 ± 0.0155
3. Gallic acid derivatives	0.0093 ± 0.0012	9.8197 ± 0.0153
6. Quercetin-3-arabinoside	11.2497 ± 0.0312	11.2647 ± 0.0222
7. Apigenin	22.0347 ± 0.0322	13.2238 ± 0.0113
**Polyphenolcarboxylic acids (mg/mL)**		
4. Caffeic acid	5.3001 ± 0.0201	0.0028 ± 0.0010
5. Rosmarinic acid	25.1654 ± 0.0412	19.333 ± 0.0313
**Iridoids (mg/mL)**		
8. Harpagoside	1.0231 ± 0.0152	0.0093 ± 0.0021
9. Loganin	0.0254 ± 0.0031	0.0145 ± 0.0013

* Values represent the mean of triplicate quantification ± standard deviation.

**Table 3 microorganisms-10-01943-t003:** Fractional inhibitory concentration index (FICI) of the *P. longifolia* extracts/penicillin combination against the prx-MRSA-2018 strain.

MIC (mg/mL)						
Strain	Alone		In Combination			
	PlLm Extract	Penicillin	PlLm Extract	Penicillin	FICI	Interaction
**MRSA**			0.0003	2	2	Indifference
			0.0003	1	1	Indifference
			0.001	0.5	0.5	Additivity
			0.009	0.25	0.27	Synergy
			0.019	0.125	0.18	Synergy
	0.312	1	0.039	0.062	0.18	Synergy
			0.078	0.031	0.28	Synergy
			0.156	0.015	0.51	Additivity
	**PlLe Extract**	**Penicillin**	**PlLe Extract**	**Penicillin**	**FICI**	**Interaction**
			0.001	2	2	Indifference
			0.002	1	1	Indifference
			0.002	0.5	0.5	Additivity
	0.624	1	0.019	0.25	0.28	Synergy
			0.039	0.125	0.18	Synergy
			0.078	0.06	0.18	Synergy
			0.015	0.03	0.28	Synergy
			0.312	0.01	0.51	Additivity

MIC: minimum inhibitory concentration; synergy = FICI ≤ 0.5; additivity = 0.5 < FICI ≤ 1; indifference = 1 < FICI ≤ 4; the values are the mean for at least three replicates.

## Data Availability

Not applicable.

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
