# Peer review of "New Insights into the Antimicrobial Potential of Polyalthia longifolia—Antibiofilm Activity and Synergistic Effect in Combination with Penicillin against Staphylococcus aureus"

_microorganisms, 2022, doi:10.3390/microorganisms10101943_

Round 1
Reviewer 1 Report
Dear authors,
I decided to write my comments at the same time I was reading the manuscript. So, please, before deciding towards alterations, read all the comments.
A possible publication will imply major improvements are done in the structure, results’ presentation and discussion of results. Most of the times, the way we present our findings (the history we tell) changes the way the readers will receive the work (as well as reviewers). And, from my point of view, the manuscript can be scientifically more “mature”. I have some concerns about the novelty of the work; however, I think science is much more than novelty: is a way to improve humanity as a whole and should not depend greatly on the country geography and resources. That being said, if the manuscript is improved, I will agree with its publication.
Line 72-75 – The authors state that this has not been done with these two species from Cameroon. It is not clear if the novelty is the country or the species. Please clarify. The following sentence adds to this, but I think it will better rephrase the section in order to be clear what is the novelty of the work, independently of the cultivation country. Or do you think the geography can modulate the compounds produced by these two species? If yes, this should be addressed somewhere in the paper.
Line 91- how was the solvent evaporated?
Line 92- For which maximum period was the dry residue stored?
Table 1 legend: The table reports the designation of each sample, am I right? That should be clear in the legend.
Line 97- Phytochemical analysis was performed only for P. longifolia. Why? If was it done previously, why present it again in this paper?
Line 104- List of standards is available? If yes, where?
Line 109- “Test microorganisms - Staphylococcus aureus ATCC 25923, Escherichia coli ATCC 25922 109 and Candida albicans P37037”, missing “-“ after P37037.
Line 116 to 122- The four strains were cultured in both media? MH and Sabouraud? If not, please clarify. The same for section 2.5.1.
Line 127: “The MIC values were” should be “The MIC were”
Section 2.5.1 – The temperature of incubation is missing.
Line 138-140 – Rephrase since it is confusing.
Line 150-152- “Samples were taken every hour and growth rates were determined by measuring” should be “Samples were taken every hour and growth monitored by measuring”
Section 2.6 and 2.7- Which strain of S. aureus?
M&M section – missing the number of replicates in most of the cases.
Line 179-180- Not clear how it was performed. Final volume of incubation and final concentration of cells used?
Line 188- Volume of MHB, supplement and cell suspension?
Section 2.9- Which penicillin was tested?
Section 3.1- The authors should explain why they evaluated only the extracts of P. longifolia leafs. And why only the methanol and ethanol profile were shown. In the M&M section, the authors present a table with info of extracts also from A. muricata, different plant organs and for both with water extraction. If the paper will not focus on A. muricata (as the title and M&M suggest) then the Abstract, Introduction and M&M should be rewritten accordingly.
Figure 1- The figure must be improved to meet the standards of a publication. This seems a print screen of the software and should be improved. Moreover, the mAU in the B panel is missing. Additionally, we have used a PDA, the result shown is based in which wavelength? Or is the total combined spetra?
Section 3.2- Ok. Now A. muricata extracts’ results are shown as well as water extracts. Maybe this would have been the first section of results since the results justify the focus only the leaf organic solvent extracts, as I was asking previously.
Line 235- Were all the extracts promising? Overstatement and does not imply the real results: Overall, P. longifolia extracts were more active than A. muricata for the G+ strain, whereas they were about the same for G- and yeast. This, I guess, also justifies the use of S. aureus for the following assays.
Line 236- the water leaf extract was also very active against S. aureus? The sentence applies to the extracts in general, that will include water extracts.
Line 237- “The lowest MBC 237 values were also recorded for the PlLm and PlLe”. Why the use of “also”? You did not report any result for PlLm and PlLe before this sentence. The activity of m and e extracts against G+ is similar, so it is expected that its components do not vary greatly, as seen in the analysis performed. This is the reason that you should present the HPLC results afterwards. It makes more sense. Also, although minor, the MIC of PlLe is superior to PlLm. So, the only difference between the extracts -Gallic acid derivatives observed- should not contribute greatly to the bioactivity.
Line 139- Missing the table or other showing MRSA results.
Line 255- “PlLm and PlLe extracts 255 induced a significant growth delay up to 6 and 7 h compared” should read “PlLm and PlLe extracts 255 induced a significant growth delay up to 6 and 7 h, respectively, compared”?
Figure 2, Figure 3- Why present the p value in the legend if the figure does not have any reference to statistical analysis?
Line 260- Did the authors tested the bacteriostatic vs bactericidal activity with OD? Can you state the bacteriostatic activity based on OD results only? And even if OD results are analysed, do they show a bacteriostic effect? Why?
Line 269-271- “A total kill effect (no viable cells) was recorded when bacterial cells were exposed to 269 both P. longifolia tested extracts at concentrations equivalent to 2 × MIC (Figure 3)” after 2 hours of incubation. This should be clear in the same sentence. Not different sentences.
Figure 4- The document I had acess to has no images in the panel B, so I was unable to analyse it. Panel A legend is a description and does not inform exactly what is shown, which are the % of fluorescent cells (control and treated) after PI staining. I was unable to analyse the panel B properly, however, it should be evident what differs in the first column and second column for each hour. I think this panel B can be improved by providing the time analysed instead of 1, 2 and 3 and by giving a title to each column. Don’t understand why (p < 0.05) is in the legend. The mean of how many replicates? Info missing in the M&M.
Section 3.3.1- The title suggests that PlLm and PlLe extracts were tested, but only PlLm results are reported in Figure 4. As I said previously, I guess this was performed since no major differences were observed in the previous experiments and also in their composition. This must be explained in the document and in this way M&M of this sections make more sense (only focus the methanol extract). In the end of the section the authors state that PlLe extract was also evaluated. But if results are similar, why they did not show them? And why in M&M is reported that only PlLm was tested? “A membrane permeability test was performed to assess the effect of P. longifolia extract PlLm on S. aureus cell membrane integrity”
Line 289- the number of fluorescent cells significantly increase compared with the control, not over time. Indeed, the number of fluorescent cells increase over the time, but based on figure 4A, statistical analysis was performed between control and test, not over/along the time.
Figure 5- You should use the same acronymous PlLm and not write the full name. The same for S. aureus.
Section 3.3.3. – I advise the authors to present the results where they have tested both extracts before the ones where only one extract was tested. It makes more sense to the readers possibly.
Figure 6- The charts might be improved – first, the results could be presented starting with the lower concentration tested. Also, in the x-axis instead of concentration, I think it would be more informative to refer the concentration in terms of MIC. This also allow to present, if you wish, both results in the same chart. Finally, if you decide to maintain the charts separately, the bars are not centered with the concentration, and it makes the chart weird. The bars can be closer to each other.
Line 322- “and ¼ MIC (Figure 6), although the last was not significantly different”. Something like this should be stated since the effect to ¼ MIC was lower, as expected.
Line 332-333- Please uniformize the reference to extracts here and in the entire document.
Table 4- Why was PlLe extract tested at lower XMIC than PlLm? More concentrations tested for PlLe, why?
Line 351: “A. muricata and P. longifolia extracts have been described in the past to possess prominent antibacterial properties. However, their mechanisms of action are not fully understood.” – So, the novelty of the work in fact is a first insight in the MoA of extracts and their potential synergy effects with penicillin. This was not done for A. muricata. So, what was the novelty herein shown on A. muricata research that deserves its inclusion in the manuscript? Please clarify or, in alternative, rewrite your manuscript without a high emphasis on such plant in this study.
Line 365- “(MICs ranging from 0.0125 to 50 mg/mL” – typo “)” must be added.
Sentence starting line 365- Although not a big difference in the MIC, can you clarify the main differences between those already published studies, if any?
Line 369- “Regarding the activity of A. muricata extracts, we may consider that this falls within 369 the antimicrobial activity presented in previous reports for aqueous, ethanolic and meth- 370 anolic extracts obtained from different parts of the plant [29,33-35].” This information just confirms my previous comment on line 351.
Line 371- You have discussed MIC results. And what about MBC results?
Discussion of the extracts composition- The authors discuss quantities of the components, but in the results none is presented at this regard. Also, the sentences must be clearer at regard of results of the paper and comparison with previous studies.
Discussion- Authors discuss MIC, afterwards extracts, then MRSA and then MIC and MBC again. This is very confusing.
Discussion section 422-429- the authors should clarify the bacteriostatic (results) and bactericidal point of view.
Line 448- If the active compounds act on cell wall biosynthesis, for instance, could you have seen a similar effect on SEM? What I mean is: do your results give a certain response that the MoA targets the cytoplasmic membrane and causes its disruption? Or they suggest and it is an hypothesis that should be further studied with complementary techniques? Or do you have other results that support this hypothesis in your study?
Line 459- Be more accurate regarding the ¼ MIC results.
Discussion of biofilm- I miss info on the nature of the S. aureus biofilm and in which way can these compounds be acting to inhibit its formation.
Line 470- “The synergistic antimicrobial activity of P. longifolia extracts or isolated compounds (catechin, epicatechin, caffeic and rosmarinic acids) against different bacterial strains, including MRSA, was also reported by other authors”. Did some of these reports involve penicillin? Clarify this sentence. Sinergy of extracts using which antibiotic or antibiotics? Comparisons.
Sinergy discussion: Penicillin was a well-known MoA. Sinergy normally involves molecules with different targets and MoA, right? So, I think authors can enrich the discussion by presenting some discussion on such aspects.
Author Response
Response to Reviewer 1 Comments
First, the authors would like to thank the reviewer for the patient and careful evaluation of our work and for providing ideas and corrections that will improve the quality of the manuscript. We also thank the reviewer for the nice thoughts expressed at the beginning of the review, which we really appreciate.
Point 1: Line 72-75 – The authors state that this has not been done with these two species from Cameroon. It is not clear if the novelty is the country or the species. Please clarify. The following sentence adds to this, but I think it will better rephrase the section in order to be clear what is the novelty of the work, independently of the cultivation country. Or do you think the geography can modulate the compounds produced by these two species? If yes, this should be addressed somewhere in the paper.
Response 1: The novelty of the paper consists not only in the geographical origin of the plant material, although we have no doubts that the cultivation conditions influence in a significant manner the chemical composition of the plant extracts and therefore the biological properties (including antimicrobial activity). It is known that the pedo-climatic conditions impact greatly the plants’ biosynthetic capacity, inducing changes in the chemical profile of all secondary metabolites. Such metabolites are produced as a mechanism to adapt to the environmental changes, polyphenols being known as photoprotectors against radiations and acting as antioxidants to prevent free radicals’ formation, thus reducing cell membrane degradation. (BILLET, Kévin, et al. Field-based metabolomics of Vitis vinifera L. stems provides new insights for genotype discrimination and polyphenol metabolism structuring. Frontiers in plant science, 2018, 9: 798.) After a careful literature survey, we may say that less is known about the chemical composition of Polyalthia longifolia leave extracts (especially wild plants), mechanism of action, antibiofilm activity and synergistic activity with penicillin against MRSA strains. As reviewer suggested, we modified the Introduction section to present better the novelty of our work, as follows:
To the best of the author's knowledge, no studies have been carried out on the antimicrobial properties of P. longifolia leaf and stem extracts from wild plants of the Cameroon region. Although various papers have confirmed the antimicrobial activities of the crude extracts and isolated compounds from P. longifolia, their mechanism of action is not fully understood. In addition, less is known about the chemical composition (most studies contain data regarding the volatile fraction and preliminary screening of the bioactive metabolites without the indication of any polyphenolic compounds), antibiofilm activity and synergistic effect with penicillin against pathogenic microbial strains. Therefore, in the present study we address not only the antibacterial and antifungal properties of P. longifolia extracts but also their chemical composition (identification and quantification by RP-UPLC-PDA techniques), mechanism of action and the effect in combination with penicillin against a MRSA strain, as well as the activity against biofilm formation – an important S. aureus virulence factor.
Point 2: Line 91- how was the solvent evaporated?
Response 2: Organic solvents (ethanol and methanol) were evaporated under reduced pressure in a rotary-evaporator to obtain crude extracts. Water was evaporated in an oven at low temperature.
Point 3: Line 92- For which maximum period was the dry residue stored?
Response 3: The dry residue was stored one week at 4oC until the testing.
Point 4: Table 1 legend: The table reports the designation of each sample, am I right? That should be clear in the legend.
Response 4: Table 1 was removed from the manuscript. Information about the code of each sample were introduced in the legend of the new Table 1.
Point 5: Line 97- Phytochemical analysis was performed only for P. longifolia. Why? If was it done previously, why present it again in this paper?
Response 5: We performed the phytochemical analysis only for Polyalthia longifolia leaf extracts (PlL m and PlL e) based on the antimicrobial activity. The analysis was not done previously and was not reported elsewhere. All available literature contains preliminary data concerning the major groups of compounds found in various extracts obtained from this species. However, there is only general screening by identification reactions and total quantification by spectrophotometry. There is only one article that mentions the liquid chromatography, but the identification and quantifications of the peaks from the chromatograms is not included in the publication. Therefore, at the present no data is available concerning the type of polyphenols and flavonoids from this plant species.
As reviewer suggested, the structure of the manuscript was changed and all the data regarding Annona muricata were removed from the manuscript.
Point 6: Line 104- List of standards is available? If yes, where?
Response 6: Sorry for this slip, we included a list of the used standards which are found in the data library of the Chromeleon software, for which we also have established calibration curves. Following information were include in the manuscript:
The available standards were catechin, epicatechin, caffeic acid, rosmarinic acid, chlorogenic acid, luteolin, apigenin, quercetin-3-arabinoside, apigenin-7-O-glucoside, luteolin-7-O-glucoside, loganin, and harpagide (all HPLC quality, Sigma, Germany). The standard stock solutions in concentrations varying from 0.114–0.195 mg/mL were analysed in identical conditions with the samples and the calibration curves were obtained with a standards deviation of 0.009 and a correlation coefficient of 0.9989. The limit of detection (LOD) and the limit of quantification (LOQ) of epicatechin and chlorogenic acid were calculated at 280 ng/mL and 145 ng/mL.
Point 7: Line 109- “Test microorganisms - Staphylococcus aureus ATCC 25923, Escherichia coli ATCC 25922 109 and Candida albicans P37037”, missing “-“ after P37037.
Response 7: We thank the reviewer for his observation. “-“ was added in the manuscript after P37037
Point 8: Line 116 to 122- The four strains were cultured in both media? MH and Sabouraud? If not, please clarify. The same for section 2.5.1.
Response 8: To make the text clearer, following changes were made in the manuscript:
Prior to experiments, S. aureus and E. coli strains were transferred to Mueller-Hinton agar (MHA, Liofilchem, Italy) and C. albicans on Sabouraud dextrose agar (SDA, Carl Roth, Germany) and incubated at 37°C.
Section 2.5.1. - The inoculum was added to each well of a microplate (final cell densities approximately 1.5 × 106 CFU/mL for bacterial inoculum and 2.5 × 103 CFU/mL for fungal inoculum), except the wells containing only MHB (used for S. aureus and E. coli strains) or SDB (used for C. albicans strain) medium considered as blank.
Point 9: Line 127: “The MIC values were” should be “The MIC were”
Response 9: The phrase was changed, following the reviewer suggestion:
The MIC were determined by the broth microdilution method, as we previously described.
Point 10: Section 2.5.1 – The temperature of incubation is missing.
Response 10: The temperature of incubation for MIC determination was not changed from our previous reports [16,17]. However, to clarify this information for MBC, the following change was performed:
To evaluate MBC or MFC, a volume of 15 μL was taken from each well with no visible growth, inoculated on MHA or SDA plates and incubated at 37°C.
Point 11: Line 138-140 – Rephrase since it is confusing.
Response 11: At reviewer suggestions, we have reformulated the sentence as follows:
The MBC/MFC was considered the lowest concentration at which bacterial and fungal cells failed to grow after plating onto MHA or SDA plates.
Point 12: Line 150-152- “Samples were taken every hour and growth rates were determined by measuring” should be “Samples were taken every hour and growth monitored by measuring”
Response 12: The correction proposed by the reviewer suggestions was made in the manuscript:
Samples were taken every hour and growth monitored by measuring the optical density at 600 nm (OD600), using a Beckman Coulter DU 730 spectrophotometer.
Point 13: Section 2.6 and 2.7- Which strain of S. aureus?
Response 13: The full name of the strain (S. aureus cells ATCC 25923) was introduced in sections 2.6 and 2.7.
Point 14: M&M section – missing the number of replicates in most of the cases.
Response 14: The next sentence was introduced in section 2.10. Statistical analysis:
All experiments were performed in triplicate
Point 15: Line 179-180- Not clear how it was performed. Final volume of incubation and final concentration of cells used?
Response 15: We thank the reviewer for spotting the inaccuracy. We modified the sentence as follows:
A 5 mL volume of a S. aureus ATCC 25923 cell suspension (approximately 1 × 108 CFU/mL) was incubated for 4 h in the presence of PlLm extract (final concentrations equivalent to MIC and 2 × MIC values).
Point 16: Line 188- Volume of MHB, supplement and cell suspension??
Response 16: The requested information was introduced in the manuscript, as follows:
In each well, a volume of 100 μL Luria Bertani medium (LB) was inoculated with 100 μL bacterial suspensions (final density approximately 1 × 106 CFU/mL). Only in the wells considered as samples the medium was supplemented with PlLm and PlLe extracts to reach final concentrations ranging from 0.0097 to 0.312 mg/mL.
Point 17: Section 2.9 - Which penicillin was tested?
Response 17: Penicillin G sodium salt (Roth) was used for all experiments. The sentence was changed accordingly.
Point 18: Section 3.1- The authors should explain why they evaluated only the extracts of P. longifolia leafs. And why only the methanol and ethanol profile were shown. In the M&M section, the authors present a table with info of extracts also from A. muricata, different plant organs and for both with water extraction. If the paper will not focus on A. muricata (as the title and M&M suggest) then the Abstract, Introduction and M&M should be rewritten accordingly.
Response 18: The data regarding A. muricata and consequently Table 1 were removed from the manuscript. Also, we swapped sections 3.1. with 3.2 in the new version of the manuscript. Accordingly, the next sentence was change in the manuscript to explain better why the phytochemical analysis and antibacterial tests were performed only for PlLm and PlLe extracts (based on the lowest MIC and MBC values, as the reviewer noticed otherwise in his comments):
Based on lower MIC and MBC values, extracts PlLm and PlLe were selected to perform further phytochemical analysis and antibacterial tests against S. aureus strains.
Point 19: Figure 1- The figure must be improved to meet the standards of a publication. This seems a print screen of the software and should be improved. Moreover, the mAU in the B panel is missing. Additionally, we have used a PDA, the result shown is based in which wavelength? Or is the total combined spetra?
Response 19: Thank you for your indication. The figure was improved, and we ensured that all necessary elements are included and visible in the chromatograms. The included chromatograms are the images obtained at 330 nm for which the peaks are the highest.
Point 20: Section 3.2- Ok. Now A. muricata extracts’ results are shown as well as water extracts. Maybe this would have been the first section of results since the results justify the focus only the leaf organic solvent extracts, as I was asking previously.
Response 20: As we previously mentioned, A. muricata data were removed from the manuscript. Also, we swapped sections 3.1. with 3.2 to make the paper easier to understand.
Point 21: Line 235- Were all the extracts promising? Overstatement and does not imply the real results: Overall, P. longifolia extracts were more active than A. muricata for the G+ strain, whereas they were about the same for G- and yeast. This, I guess, also justifies the use of S. aureus for the following assays.
Response 21: We thank the reviewer for spotting the inaccuracy. Accordingly, the section was changed as follows:
The results of the antibacterial and antifungal activities are presented in Table 1. P. longifolia extracts showed a promising activity against S. aureus (MICs as low as 0.039 mg/mL) and were moderately active against E. coli (MICs of 5 and 10 mg/mL). The lowest MBC values were recorded for PlLm and PlLe extracts against S. aureus (0.078 mg/mL and 0.156 mg/L respectively). The same extracts also showed a good antibacterial activity against a MRSA clinical isolate, with MIC values of 0.312 and 0.624 mg/mL, respectively and MBC value of 10 mg/mL. Concerning the antifungal activity, the lowest MIC or MFC recorded value was 5 mg/L for several P. longifolia extracts.
Point 22: Line 236- the water leaf extract was also very active against S. aureus? The sentence applies to the extracts in general, that will include water extracts.
Response 22: The section was changed as we showed at point 21.
Point 23: Line 237- “The lowest MBC 237 values were also recorded for the PlLm and PlLe”. Why the use of “also”? You did not report any result for PlLm and PlLe before this sentence. The activity of m and e extracts against G+ is similar, so it is expected that its components do not vary greatly, as seen in the analysis performed. This is the reason that you should present the HPLC results afterwards. It makes more sense. Also, although minor, the MIC of PlLe is superior to PlLm. So, the only difference between the extracts -Gallic acid derivatives observed- should not contribute greatly to the bioactivity.
Response 23: The sentence was changed following the reviewer comments:
The lowest MBC values were recorded for PlLm and PlLe extracts against S. aureus (0.078 mg/mL and 0.156 mg/L respectively).
The data regarding the phytochemical analysis were moved afterwards. Indeed, PlLm and PlLe extracts were chosen based on their antibacterial activity.
Point 24: Line 139- Missing the table or other showing MRSA results.
Response 24: We chosen to present the values for MIC and MBC against the MRSA strain in the text and not in the table from practical reasons (the test was performed only for selected P. longifolia extracts).
Point 25: Line 255- “PlLm and PlLe extracts 255 induced a significant growth delay up to 6 and 7 h compared” should read “PlLm and PlLe extracts 255 induced a significant growth delay up to 6 and 7 h, respectively, compared”?
Response 25: The sentence was modified as reviewer suggested – Line 298.
Point 26: Figure 2, Figure 3- Why present the p value in the legend if the figure does not have any reference to statistical analysis?
Response 26: The p value was removed from the legend
Point 27: Line 260- Did the authors tested the bacteriostatic vs bactericidal activity with OD? Can you state the bacteriostatic activity based on OD results only? And even if OD results are analysed, do they show a bacteriostic effect? Why?
Response 27: The experiments regarding the growth inhibition were performed based only on OD measurements. We did not assess the growth inhibition by plating and CFU counting, as we did for the time kill assay. As Figure 2 shows, no modification of the ODs for MIC and 2xMIC were recorded during the entire experiment, therefore we presumed a bacteriostatic effect.
Point 28: Line 269-271- “A total kill effect (no viable cells) was recorded when bacterial cells were exposed to both P. longifolia tested extracts at concentrations equivalent to 2 × MIC (Figure 3)” after 2 hours of incubation. This should be clear in the same sentence. Not different sentences.
Response 28: The section was change, as reviewer suggested:
A total kill effect (no viable cells) was recorded when bacterial cells were exposed to both P. longifolia tested extracts at concentrations equivalent to 2 × MIC after only 2 h of incubation (Figure 3).
Point 29: Figure 4- The document I had acess to has no images in the panel B, so I was unable to analyse it. Panel A legend is a description and does not inform exactly what is shown, which are the % of fluorescent cells (control and treated) after PI staining. I was unable to analyse the panel B properly, however, it should be evident what differs in the first column and second column for each hour. I think this panel B can be improved by providing the time analysed instead of 1, 2 and 3 and by giving a title to each column. Don’t understand why (p < 0.05) is in the legend. The mean of how many replicates? Info missing in the M&M.
Response 29: We are sorry about the situation. Following the reviewer suggestion, the legend of Figure 4 was changed. Also, we replaced 1b, 2b and 3b with Control, PlLm (15 min) and PlLm (2 h). The data are the mean of three replicates as we mentioned in section 2.10. Statistical analysis (revised form). To avoid further technical issues, we provide bellow the new version of Figure 4.
Figure 4. Effect of P. longifolia methanolic leaf extracts exposure on S. aureus cell membrane integrity. Exponential-phase cells were treated with DMSO (control), PlLm extract (concentration equivalent to ½ MIC - samples) and stained with propidium iodide. Negligible levels of fluorescence were detected in control after 3 h of incubation. Red fluorescent cells were detected in samples after 15 min and 2 h of incubation with PlLm indicating membrane damage (a). The number of fluorescent cells significantly increased over time in PlLm exposed samples (b). Values are the mean ± S.E.M. Bars indicate standard deviations. Asterisks denote a significant difference vs. Control (*** = p = 0.0001; **** = p < 0.0001).
Point 30: Section 3.3.1- The title suggests that PlLm and PlLe extracts were tested, but only PlLm results are reported in Figure 4. As I said previously, I guess this was performed since no major differences were observed in the previous experiments and also in their composition. This must be explained in the document and in this way M&M of this sections make more sense (only focus the methanol extract). In the end of the section the authors state that PlLe extract was also evaluated. But if results are similar, why they did not show them? And why in M&M is reported that only PlLm was tested? “A membrane permeability test was performed to assess the effect of P. longifolia extract PlLm on S. aureus cell membrane integrity”
Response 30: Indeed, we performed this test with both extracts. The results were very similar so, to avoid the redundancy we have chosen to present in Figure 4 only the results for one extract (PlLm). To avoid misunderstanding, section 2.6. Evaluation of membrane integrity was changed as follows:
A membrane permeability test was performed to assess the effect of PlLm and PlLe extracts on S. aureus ATCC 25923 cell membrane integrity.
Point 31: Line 289- the number of fluorescent cells significantly increase compared with the control, not over time. Indeed, the number of fluorescent cells increase over the time, but based on figure 4A, statistical analysis was performed between control and test, not over/along the time.
Response 31: We thank the reviewer for proposed correction. We changed this section as follows:
Following exposure to PlLm extract, the number of fluorescent cells significantly increased compared with the control, starting with 15 minutes of incubation (p = 0.0001).
Point 32: Figure 5- You should use the same acronymous PlLm and not write the full name. The same for S. aureus.
Response 32: The legend of Figure 5 was changed, following the reviewer suggestion:
Figure 5. SEM photomicrographs showing the effects of PlLm extract on S. aureus cell morphology: a - control; b, c - cells exposed for 4 h to MIC and 2 × MIC, respectively. White arrows indicate irreversible morphological damage of treated bacterial cells and cellular debris.
Point 33: Section 3.3.3. – I advise the authors to present the results where they have tested both extracts before the ones where only one extract was tested. It makes more sense to the readers possibly.
Response 33: We thank the reviewer for his advice. Section 3.3.3. was modified into Section 3.4. Bacterial biofilm formation was inhibited by PlLm and PlLe extracts. Also, we modified the structure of the manuscript in accordance with all reviewers' requests.
Point 34: Figure 6- The charts might be improved – first, the results could be presented starting with the lower concentration tested. Also, in the x-axis instead of concentration, I think it would be more informative to refer the concentration in terms of MIC. This also allow to present, if you wish, both results in the same chart. Finally, if you decide to maintain the charts separately, the bars are not centered with the concentration, and it makes the chart weird. The bars can be closer to each other.
Response 34: We thank for the suggestions. Figure 6 was modified considering the reviewer comments:
Point 35: Line 322- “and ¼ MIC (Figure 6), although the last was not significantly different”. Something like this should be stated since the effect to ¼ MIC was lower, as expected.
Response 35: Following the reviewer suggestions, the sentence was changed as follows:
We must emphasize that the antibiofilm activity was also recorded for subinhibitory concentrations such as ½ MIC and ¼ MIC (Figure 6), although the last concentration was not significantly different compared with the control.
Point 36: Line 332-333- Please uniformize the reference to extracts here and in the entire document.
Response 36: Where possible, all required modifications were made in the manuscript.
Point 37: Table 4- Why was PlLe extract tested at lower XMIC than PlLm? More concentrations tested for PlLe, why?
Response 37: Table 4 was modified by adding new concentrations for PlLm, as reviewer suggested.
Point 38: Line 351: “A. muricata and P. longifolia extracts have been described in the past to possess prominent antibacterial properties. However, their mechanisms of action are not fully understood.” – So, the novelty of the work in fact is a first insight in the MoA of extracts and their potential synergy effects with penicillin. This was not done for A. muricata. So, what was the novelty herein shown on A. muricata research that deserves its inclusion in the manuscript? Please clarify or, in alternative, rewrite your manuscript without a high emphasis on such plant in this study.
Response 38: As we previously presented in our response, A. muricata data were removed from the paper and consequently the manuscript re-written.
Point 39: Line 365- “(MICs ranging from 0.0125 to 50 mg/mL” – typo “)” must be added.
Response 39: The correction was made in the manuscript, thank you for spotting it.
Point 40: Sentence starting line 365- Although not a big difference in the MIC, can you clarify the main differences between those already published studies, if any?
Response 40: Following the reviewer suggestion, we modified the sentence:
Additionally, P. longifolia extracts presented a more pronounced antifungal activity (MICs of 5 or 10 mg/mL) compared with crude leaf extracts previously tested (MIC = 25 mg/mL).
Point 41: Line 369- “Regarding the activity of A. muricata extracts, we may consider that this falls within 369 the antimicrobial activity presented in previous reports for aqueous, ethanolic and meth- 370 anolic extracts obtained from different parts of the plant [29,33-35].” This information just confirms my previous comment on line 351.
Response 41: The sentence was removed from the manuscript.
Point 42: Line 371- You have discussed MIC results. And what about MBC results?
Response 42: We thank the reviewer for spotting this error. We added comments about the MBC results:
PlLm and PlLe showed the lowest MBC values from all tested extracts - 0.078 and 0.156 mg/mL respectively. The recorded MBC values are comparable to some previous reported data for ethanolic extracts (MBC range between 0.02 and 2.5 mg/ml or diterpenoid isolated from P. longifolia var. pendula (MBC = 0.185 mg/mL), altghough there is lim-ited existing literature on P. longifolia extracts bactericidal activity.
Point 43: Discussion of the extracts composition- The authors discuss quantities of the components, but in the results none is presented at this regard. Also, the sentences must be clearer at regard of results of the paper and comparison with previous studies.
Response 43: New experiments concerning identification and quantification by RP-UPLC-PDA techniques of the extracts were performed and consequently Table 1 was modified. The discussion section was also changed.
Point 44: Discussion- Authors discuss MIC, afterwards extracts, then MRSA and then MIC and MBC again. This is very confusing.
Response 44: The discussion section was re-written, according to the new structure of the manuscript. Consecutively the order of the paragraphs was changed, hopefully in a more comprehensible manner.
Point 45: Discussion section 422-429- the authors should clarify the bacteriostatic (results) and bactericidal point of view.
Response 45: the next sentences were modified in the above-mentioned section:
Increasing the concentration up to MIC and 2 × MIC progressively inhibited S. aureus growth over time, denoting a significant dose-dependent inhibitory effect. We must emphasize that no turbidity was recorded for all samples exposed to concentrations equivalent to MIC and 2 × MIC during the 12 h experiments, implying important bacteriostatic activity.
Point 46: Line 448- If the active compounds act on cell wall biosynthesis, for instance, could you have seen a similar effect on SEM? What I mean is: do your results give a certain response that the MoA targets the cytoplasmic membrane and causes its disruption? Or they suggest and it is an hypothesis that should be further studied with complementary techniques? Or do you have other results that support this hypothesis in your study?
Response 46:
Our results clearly showed that after PlLm and PlLe exposure the cellular membrane was permeabilized for PI, implying a membrane mechanism of action. SEM was employed to check for morphological damages and cell lysis, due to the impairment of membrane integrity. However, compounds such as catechin identified in the tested extracts could interfere with the cell wall synthesis. Therefore, we do not exclude the hypothesis of multiple cellular targets for PlLm and PlLe, but we do not have sufficient experimental evidences at this moment. Consequently, we change the manuscript as follows:
Therefore, SEM was employed to determine morphological damages and the possible cell lysis induced by P. longifolia leaf extract exposure. SEM image analysis showed severe morphological alterations, extensive cell shrinkage and agglutination, with cellular debris most likely resulted from cell lysis. All these results clearly indicate that the P. longifolia leaf extracts target the cellular membrane, inducing membrane disruption and cell lysis. Our hypothesis concerning the mechanism of action is supported by previous reports showing that compounds such as catechin, identified in both PlLm and PlLe extracts, exert antibacterial effect by intercalating into the lipid bilayer leading to lateral expansion and membrane disruption. In addition, other compounds detected in P. longifolia leaf extracts such as apigenin, rosmarinic acid and caffeic acid also target the cellular membrane, as previously showed. On the other hand, catechin is known to interfere with the peptidoglycan biosynthesis. Therefore, we may presume that PlLm and PlLe extracts could act on multiple cellular targets – membrane and cell wall, but further studies are necessary to clarify this hypothesis.
Point 47: Line 459- Be more accurate regarding the ¼ MIC results.
Response 47: The sentence was modified according to the statistical analysis results, as follows:
Moreover, the inhibition effect was recorded for sub-MIC levels (½ MIC), denoting an important antibiofilm activity.
Point 48: Discussion of biofilm- I miss info on the nature of the S. aureus biofilm and in which way can these compounds be acting to inhibit its formation.
Response 48: Additional information were added to the section:
The inhibition of biofilm formation could be the result of the EPS structural components degradation or the inhibition in the DNA and RNA content of the sessile cells induced by compounds such as catechin.
Point 49: Line 470- “The synergistic antimicrobial activity of P. longifolia extracts or isolated compounds (catechin, epicatechin, caffeic and rosmarinic acids) against different bacterial strains, including MRSA, was also reported by other authors”. Did some of these reports involve penicillin? Clarify this sentence. Sinergy of extracts using which antibiotic or antibiotics? Comparisons.
Response 49: None of the above reports involved penicillin. Accordingly, the section was changed:
The synergism between P. longifolia extracts or isolated compounds (catechin, epicatechin, caffeic and rosmarinic acids) and different antibiotics such as ampicillin, oxacillin, tetracycline, daptomycin and linezolid, erythromycin, clindamycin, cefoxitin and vancomycin was previously reported. However, less is known about the synergistic interaction of P. longifolia extracts with penicillin against MRSA strains.
Point 50: Sinergy discussion: Penicillin was a well-known MoA. Sinergy normally involves molecules with different targets and MoA, right? So, I think authors can enrich the discussion by presenting some discussion on such aspects.
Response 50: We thank the reviewer for his suggestions. We added new information to this section:
We may presume that this synergistic effect could be explained by the different mechanism of action of the molecules involved in combination: penicillin interfere with the peptidoglycan biosynthesis and therefore targets the formation of a new cell wall while PlLm and PlLe extracts target the cellular membrane by impairing its integrity.

Reviewer 2 Report
The demand for new substances for treatment and prevention of bacterial infections is increasing with the worldwide spread of acquired antimicrobial resistance among the microorganisms. This is especially important for healthcare-associated infections, exposed to a significant selective pressure of a closed system of in-patients organizations and pathogens’ propensity of to the of resistance factors transmission. The work by Mihaela Savu et al is of topical issue in the field and can be interesting for the specialists. I recommend it for the publication after the introduction of several corrections and clarifications.
There are some inconsistencies in the text concerning the experiment workflow: in Materials and Methods section authors describe the aqueous extracts of Annona muricata and Polyalthia longifolia, however, there are no further mentions in the Results or Discussion. Please, clarify.
It is also not clear why the authors describe the antimicrobial activity of A. muricata if it was not further included in the study. You haven't done an extensive screening of extracts, and this data is just a distraction. I propose to remove it from the text. Alternatively, I recommend to swap the Results parts "3.1. The chemical composition of PlLm and PlLe extracts" and "3.2. Antimicrobial activity of P. longifolia and A. muricata extracts".
It would be interesting to discuss why there are such differences in MIC and MBC for different parts extracts? Is this due to difference in composition or components concentration in extracts. Why did you provide chemical composition analysis for Polyalthia longifolia leaf extracts only?
Unfortunately, it is not entirely evident from the text what new aspects of P. longifolia antimicrobial properties the authors found out, since, much is already known from the literature, as follows from the introduction and discussion (Ref 29,33-35, 17-20 etc). Please, emphasize the novelty of your work
Author Response
Response to Reviewer 2 Comments
First, the authors would like to thank the reviewer for the patient and careful evaluation of our work and for providing corrections and clarifications that will improve the quality of the manuscript.
Point 1: There are some inconsistencies in the text concerning the experiment workflow: in Materials and Methods section authors describe the aqueous extracts of Annona muricata and Polyalthia longifolia, however, there are no further mentions in the Results or Discussion. Please, clarify.
Response 1: The data for Annona muricata were removed from the text and consequently the manuscript was re-written. Also comments about Polyalthia longifolia aqueous extracts were introduced in the manuscript:
No important differences were registered between the antimicrobial activity of methanolic and ethanolic leave and stem extracts, most probably due to the similar chemical composition. However, the MIC and MBC values recorded for water extracts against S. aureus ATCC 25923 were significantly higher compared with methanolic and ethanolic extracts, and this could be related to the different solvent used for extraction.
Point 2: It is also not clear why the authors describe the antimicrobial activity of A. muricata if it was not further included in the study. You haven't done an extensive screening of extracts, and this data is just a distraction. I propose to remove it from the text. Alternatively, I recommend to swap the Results parts "3.1. The chemical composition of PlLm and PlLe extracts" and "3.2. Antimicrobial activity of P. longifolia and A. muricata extracts".
Response 2: As reviewer suggested, the data for Annona muricata were removed from the manuscript. Also, we swapped sections 3.1. with 3.2 in the new version of the manuscript.
Point 3: It would be interesting to discuss why there are such differences in MIC and MBC for different parts extracts? Is this due to difference in composition or components concentration in extracts. Why did you provide chemical composition analysis for Polyalthia longifolia leaf extracts only?
Response 3:
We have chosen extracts PlLm and PlLe to perform further tests based on the lower MIC and MBC values against S. aureus. To make the text clearer, the following change was performed in the manuscript:
Based on lower MIC and MBC values, extracts PlLm and PlLe were selected to perform further phytochemical analysis and antibacterial tests against S. aureus strains.
Point 4: Unfortunately, it is not entirely evident from the text what new aspects of P. longifolia antimicrobial properties the authors found out, since, much is already known from the literature, as follows from the introduction and discussion (Ref 29,33-35, 17-20 etc). Please, emphasize the novelty of your work
Response 4: The novelty of the work was highlighted in several sections of the manuscript:
To the best of the author's knowledge, no studies have been carried out on the antimicrobial properties of P. longifolia leaf and stem extracts from wild plants of the Cameroon region. Although various papers have confirmed the antimicrobial activities of the crude extracts and isolated compounds from P. longifolia, their mechanism of action is not fully understood. In addition, less is known about the chemical composition (most studies contain data regarding the volatile fraction and preliminary screening of the bioactive metabolites without the indication of any polyphenolic compounds), antibiofilm activity and synergistic effect with penicillin against pathogenic microbial strains. Therefore, in the present study we address not only the antibacterial and anti-fungal properties of P. longifolia extracts but also their chemical composition (identification and quantification by RP-UPLC-PDA techniques), mechanism of action and the effect in combination with penicillin against a MRSA strain, as well as the activity against biofilm formation – an important S. aureus virulence factor
